# Long-Term Adoption or Abandonment of Smart Technology in the Chinese Elderly Home Care Environment: A Qualitative Research Study

**DOI:** 10.3390/healthcare11172440

**Published:** 2023-08-31

**Authors:** Jiahao Yu, Jianyuan Huang, Qi Yang

**Affiliations:** 1Population Research Institute, Hohai University, Nanjing 211100, China; galeyhhu@hhu.edu.cn; 2Population Research Institute, Nanjing University of Posts and Telecommunications, Nanjing 210042, China; yangqi1992@njupt.edu.cn

**Keywords:** home care, older adult, smart technology, technology acceptance models, qualitative study

## Abstract

China’s rapidly aging population and shortage of care resources have made it difficult for its traditional model to meet the home care needs of the elderly. On this premise, China is implementing home digital health interventions based on smart technology. During implementation, instead of the expected explosion in long-term adoption, there has been a large amount of abandonment. But so far, the relationship between service experience and these behaviors has been ignored. This study aims to explore the reasons for the long-term adoption or abandonment behaviors of technology by elders in the home care environment. A qualitative study was conducted based on Golant’s framework of smart technology adoption behaviors among elders. Semi-structured interviews were conducted with 26 elders who are long-term or former users of smart technology in a home care environment, and data from the interviews were analyzed using directed content analysis. This study identified three themes that influence elders’ adoption behaviors of smart technology in the home care environment, including immediate effectiveness, long-term usability, and possible collateral damage. The findings indicated that the experience of the elders is the key point that affects long-term adoption behavior. For more elders to use smart technology in the home care environment, it is necessary for the government, technology developers, and nursing institutions to further reform the existing system.

## 1. Introduction

As one of the most populous countries in the world, China reported in its seventh national census bulletin that by the end of 2020, the population aged 65 years and older reached 190.64 million (13.5% of the total population), with an elderly dependency ratio of 19.8% [1]. Affected by the expansion of the phenomenon of illness and disability, the number of disabled and semi-disabled elderly people in need of professional healthcare services in China has also increased dramatically. One study reported that more than 76% of the Chinese elderly suffer from at least one chronic disease, such as diabetes, hypertension, coronary heart disease, or respiratory disease [2]. This is the “cost of victory” that comes with increasing life expectancy [3]. Additionally, influenced by the cultural philosophy of “xiao” and the long-standing one-child policy, most current Chinese families need to care for more than four older adults (both parents and even grandparents of the couple) [4]. This far exceeds the caregiving capacity of family members. Thus, the rapidly growing elderly care needs pose a great challenge to the traditional Chinese home care model that relies on family members and community agencies to provide services.

Against this backdrop, China has focused on the integration of the internet, IoT, and home care. It tries to use smart technology to act as an intermediary between institutions and the elderly to interconnect information between homes, elderly care service companies, and medical institutions [5]. China has launched “digital health interventions” such as shared nurses [6], internet hospitals [7], and smart care service platforms to enable the elderly to receive professional care services at home. However, a large number of Chinese studies are reporting that the quality and actual usage of these services is not as high as expected [8]. Even with increasing investment from the government and the market, only a small percentage of the elderly are using the technology in the long term.

With the development of technology and concepts, the scope of smart technology used to help the elderly better cope with aging and have a healthier and independent life is constantly expanding. Based on recent research, intelligent technology has become a general term for several technologies. It mainly includes (1) information technology services to help the elderly record their health status, provide telemedicine intervention, and make emergency calls [9,10]; (2) wearable devices and IoT devices that collect physical signs and home environment data of the elderly based on sensor technology [11,12,13]; (3) robot technology to help the elderly live at home [14], and (4) data processing technologies, such as smart eldercare platforms, cloud computing, and big data, that support the above products (or services) [8,15]. The remarkable feature of these technologies is that with the participation of network systems and sensors, their information collection, communications, and responses rarely require manual operation by the elderly Unless there are some specific circumstances, such as measuring blood pressure and other operations that require manual participation, or the elderly contact medical service institutions to provide door-to-door services, the device will automatically send information deviating from the preset data and provide an alarm.

In addition, smart technology has been widely demonstrated to improve individual health outcomes when integrated with healthcare systems [16]. In particular, it can improve perceived health status and safety, reduce social isolation [17], improve the overall quality of life, reduce the number of doctor visits, frequency, and length of hospital stays [18], and provide access to long-term care at home for elders with chronic illness or disability [19,20]. However, from a global perspective, only a small number of elders are aware of the new possibilities that smart products offer to improve their lives in old age, while even fewer will use them.

While recent surveys have reported a large increase in the level of technology use among elders, it is still basically focused on applications in areas such as social networking and news [21]. Only a small number of elders rely on smart technology to access home care services. From past research experience, the barriers to the adoption of smart technologies for the elderly to access care services are more related to subjective feelings and judgments during the use process, in addition to the digital divide and sociodemographic factors [22]. On this basis, some researchers have analyzed the factors influencing the adoption of smart technologies by the elderly to improve home care based on theories such as the technology acceptance model (TAM) and the unified theory of acceptance and use of technology models (UTAUT and UTAUT2) [23,24,25]. They found that adoption behavior was related to the elder’s perceptions of the usefulness of the technology, expected improvements in the efficiency and quality of care, and advantages compared to traditional care acquisition procedures [26,27]. In the meantime, it was related to usability and accessibility when using the technology; for example, ease of understanding and learning, convenience, and usability of the technical support when using it [28,29]. In addition, a previous study also reported that elderly individuals are worried about data privacy and security problems [30]. These concerns focus on potential medical risks, digital surveillance, and infringement on traditional social behavior [31,32].

However, most of these studies still focus on the influence of technical characteristics on the adoption behavior of the elderly. They think that technical threshold factors, such as ignorance of technology and the inability to acquire guidance from past experiences, are important factors affecting the adoption of technology by the elderly [33,34]. Some researchers suggest that the government should strengthen publicity to increase the trust of the elderly in technology and ask developers to improve the usability of technology [35]. However, the essence of intelligent technology in a home care environment is to place the elderly at the center of the care network [36]. Let the elderly monitor their health and seek professional help in times when necessary. This is a just-needed service for the elderly. Unlike non-just-needed technologies, such as digital social networking, they pay more attention to the practical value generated by smart technology. It means that we should not only consider the technology adoption characteristics of the elderly before they use it, but also consider the experience during the use process. Therefore, the purpose of this study is to understand and describe the service experience in the process of adopting and using smart technology in the home care environment of the elderly and to explore the influencing factors of their long-term adoption or abandonment of the technology.

## 2. Methods

### 2.1. Research Design

This study used a qualitative analysis design. We used Golant’s theory of elders’ smart technology adoption behavior as the central chain of inquiry. A semi-structured interview approach focused on collecting opinions about the long-term adoption or abandonment of smart technology by the elderly in the home care environment. In order to ensure the interviewee’s right to know, they were informed of the purpose, method, and usage of the research. The corresponding data were kept confidential, and sensitive information was excluded. The appropriate procedures for this study were approved by the Academic Review Committee of the first author’s university. All participants participated in the survey voluntarily and signed a written informed consent form.

Golant’s (2017) theory of the elder’s smart technology adoption behaviors [37] was used to guide the investigation and content analysis of this research. This theory holds that the coping process of the elderly in the face of smart technology begins with unsatisfied needs and is influenced by factors such as perceived effectiveness (usefulness, relative advantage), usability (ease of use), and possible collateral damage in the decision-making process. This theory also puts forward that the acceptance behavior of the elderly is not a static and time-point decision-making event but a long-term investigation process. The relationship between the elderly and smart technology is not a simple either/or decision. In the process of seeking to meet the needs of the elderly, they may adopt a series of mixed ways as a coping plan. It can be seen that this theory is essentially a whole process technology use model, which can not only analyze the technology adoption at a certain point in time but also be applicable to the analysis of long-term technology adoption or abandonment behavior.

### 2.2. Sampling and Recruitment

This study was conducted in Nanjing, in Eastern China. As one of the first pilot regions in China for the application of smart elderly care, it has much experience accumulated in the process of practice. To help the elderly live at home better, besides personal purchases, the Nanjing government promotes the use of smart technology in the elderly home care environment through the form of government purchases. The government provides the necessary devices (such as smart bracelets, low-value sensors, or emergency buttons) in the form of welfare. At the same time, it is equipped with free door-to-door services (such as professional care and emergency visits) three times a month. The elderly can obtain services through automatic feedback from devices or active calls by themselves. After receiving the information, the smart platform will match the nearby healthcare staff to visit.

The elderly who participated in the study were recruited through local health management departments and community health service centers. Our recruitment criteria were: (1) people over 60 choose to grow old in a family environment; (2) being sick or existing to meet the care needs (those suffering from acute and critical diseases were excluded); (3) they have the ability of independent judgment and can communicate normally (those who were severely disabled, demented, or with aphasia were excluded); and (4) smart technology was being or has been used in the family environment. To ensure that enough information was collected, recruitment was a long-term process until no new topics emerged.

### 2.3. Pre-Data Collection

Based on building a systematic understanding of existing relevant research, we conducted an extensive discussion based on the selected theoretical framework to form an initial interview outline. The outline was then presented for review to one professional gerontologist and one enterprise manager operating a smart technology-based home care service. At the same time, two participants were selected for pre-interviews. We revised the formal outline according to the experts’ suggestions and pre-interview results.

### 2.4. Data Collection Procedure

All interviews were conducted between August and November 2022. We negotiated the specific interview time with the participant in advance. Depending on the wishes of the participant, we chose to interview in the community meeting room (*n* = 11) or their home (*n* = 15). The interviews were mainly conducted by the first author and the second author. The second author is a gerontologist with rich interview experience, and he led the whole interview process. With his efforts, we established a good relationship with the participant. During the interview, the researchers mainly listened to the participant’s opinions and observed their body language and facial expressions. The researcher did not interrupt the participant unless the researcher needed them to clarify their point or the participant elaborated for a long time on points that were not related to the study. Each interview lasted approximately 36 min (30–70 min). Field notes were recorded during the interviews and audio-recorded with the participant’s consent.

### 2.5. Data Analysis

Within 24 h of each interview, the third author independently transcribed the audio recordings, and all qualitative data were managed with Microsoft Word software. The research team worked together to analyze the data and discuss controversial points. Because this study followed Golant’s theory framework, the data collected were more structured than other qualitative studies, and the process of analysis was relatively clear [38]. Therefore, we used a directed content analysis method [39]. The specific steps included the following: (1) Defining the unit of analysis. Sentences expressing behaviors related to the long-term adoption or abandonment of smart technology for the elderly in the home care environment were defined as minimum segmentation units. (2) The original data were reviewed and read repeatedly. (3) Establishing specific categories of analysis units based on the theoretical framework. (4) The content was coded and classified. Important ideas and concepts were adopted and coded, and codes with similar content were grouped into the same category, forming themes and subthemes. (5) The results were interpreted and analyzed.

### 2.6. Rigor

This study adopts the general standards (credibility, confirmability, dependability, and transferability) proposed by Guba and Lincoln to test rigor [40]. In order to enhance credibility and confirmability, we established a good relationship with the participants and triangulated the qualitative materials in the process of transcription, analysis, and induction. To ensure the dependability of the study, we carefully designed the study and invited two experts outside the team to review it. To establish the transferability, we strictly followed the consolidated criteria for reporting qualitative research (COREQ checklist) [41] and recorded the research process, sampling methods, and the basic information of the final participants in detail (Appendix A).

## 3. Results

### 3.1. Participant Information

Table 1 provides the demographic characteristics of the participants. A total of 26 elders participated in the survey, with an average age of 71 (SD = 6.7), and more than half of them were males (*n* = 15). A total of 21 participants had received basic education (middle school and above). Almost all participants had more than one chronic disease. Among them, 6 participants were in good health and only needed daily life care, such as falling prevention; 11 participants required skilled care at home; and another 9 participants required long-term health monitoring services at home, as prescribed by their physicians. Additionally, 14 participants gave up the use of smart technology in the home care setting.

Most of the participants learned about smart technology through the advice of external experts and family members. Among them, 10 participants used smart technology under the advice of healthcare staff, 8 participants learned about it from family members and friends, and 5 participants experienced smart technology through publicity stations set up in community centers. In the use of smart technology, all participants have access to the smart eldercare platform in their homes, and most of them have installed emergency buttons. Seventeen participants used wearable or IoT devices, for example, smart bracelets, smart body monitoring devices, and smart home devices, such as fall prevention. Only five participants used partner or nursing robots.

### 3.2. Themes

The following three themes were identified in the elders’ long-term adoption and abandonment behaviors of smart technology in the home care environment: (1) the direct effectiveness, (2) the long-term usability, and (3) possible collateral damage.

#### 3.2.1. Theme 1: Direct Effectiveness

The elder’s perceptions of the effectiveness of smart technology in the home care environment are a comprehensive evaluation that is conducted on an ongoing basis throughout the service. They will continually compare their direct feelings after using smart technology with their past life experience, especially whether the technology meets their needs and whether it is more advantageous compared to traditional ways.

##### Standardized Management

Compared with traditional ways, one of the advantages of smart technology is that it can integrate previously scattered information through the platform. After calculation and processing, this information forms a complete data chain. After using smart technology, participants think that the workflow of nursing staff employed by the elderly and their families usually lacks standardization in the traditional home care environment. There was great randomness in the time and process when nurses came to work in the past. The technology allows the elderly and their families to experience the standardization of daily management and service processes, which significantly improves their feeling of reliability.


*“All service data are recorded on the platform, and you can see the result of the appointment and the progress of the nursing service. When nurses take the order, you will view their basic information, such as photos and qualifications”. (P9, adoption)*


In addition to the information about the elderly, information about other people involved in technology and the content and the process of services will be displayed. The platform will screen those idle and qualified nurses according to big data to provide the needed services for the elderly. Elders do not have to constantly discuss the service time with the nursing staff and worry about whether nurses are qualified for the services they need.

##### Traditional Care Concept Maintenance

For the elderly, the traditional approach of going directly to a professional facility is their first choice for care services. The technology offers a way for them to connect with specialist agencies, even in their home environment. The elderly can choose online medical services through the smart platform. The platform will screen out the hospitals they often go to and the nearest hospitals through big data. The elderly can freely choose the institutions they need according to their needs. At the same time, their online and offline medical records will be synchronized to their files. This form of maintaining their traditional perception of care gives them a strong sense of trust.


*“Even if I receive services at home, as long as I choose the hospital I’m scheduled to go to, my doctor will see all my records when I revisit the hospital. Likewise, the hospital can place a long-term care order for me, so I know what services I should book at what time”. (P3, adoption)*


##### Successful Early Warning

Compared to the traditional models, smart technology has the advantage of being able to prevent risky behaviors and predict the health status of the elderly. Some elderly people have successfully avoided risks by changing their lifestyle or seeking medical treatment in time, according to the prompts in the actual use process. This experience has further strengthened their belief in the long-term use of technology.


*“It really works! These devices synchronize my blood pressure and blood sugar indicators measured at home every day [……] Once it suggested that I was at risk of stroke, and I went to the hospital and already had signs of transient ischemic attack”. (P11, adoption)*


##### Speed of Response to Needs

Many elders have raised their concerns or accusations about smart technology. The most mentioned thing is devices need to interact with the smart platform through network systems. It means that if there are problems, such as device malfunction or network disruption, the help-seeking information cannot be transmitted to the platform in time. In addition, although the elderly can turn to the platform for help through smart devices such as sensors, a lot of information still needs secondary processing by humans. After receiving the information, the platform staff will match the medical institutions in the region; however, it is not always possible to find a suitable institution smoothly. This makes the elderly distrust the communication and service mode referred by technical intermediaries. Some seniors failed to receive help in critical situations. This negative experience accumulated and eventually drove them to abandon the continued use of smart technology.


*“There is an online communication function in the platform, but it tends to reply very slowly. It is very different from being in an institution, where I can always find a staff if I need them”. (P23, abandon)*


These experiences made them feel that they could not obtain a timely response to their questions in their daily lives from the equipment or callers. This service model is significantly weaker in terms of its continuity of service than traditional forms of service.

#### 3.2.2. Theme 2: Long-Term Usability

Compared with effectiveness, usability comes more from whether the elderly think technology is easy to use in the long-term process, whether they can be linked to the desired service through technology, and whether they can afford the long-term use cost.

##### Ease of Use

In the past, it was often assumed that the elders’ use of technology would be hampered by their declining physical functions (especially visual and auditory functions). However, many participants have come to believe that existing technology has made good enough technological improvements that they can easily use it.


*“These devices have amplified the words on it and raised the volume of the voice, so I can receive the information it gives effortlessly”. (P5, adoption)*


An introduction page of a smart terminal used to control the home environment shows that through training a large amount of data on the life of elders, the manufacturer had screened out several functions commonly used (like temperature adjustment, safety protection, and emergency help-seeking). These functions help the elderly to meet their needs quickly through physical buttons, while other functions interact through the touch screen of the device.


*“Most of these devices are designed very simple, with only a few buttons [……] I labeled the buttons so that you can just press them when you need them”. (P6, adoption)*


##### Nonsensitized Monitoring

Some elders showed the promotional materials of smart devices they used. We find that it usually says that devices can enter your life through technical means without any awareness. For example, a brochure of an anti-fall detector states that it adopts millimeter wave technology, which can sense human behavior through radar and automatically gives an alarm to the platform when the elderly fall. Compared with traditional video surveillance, devices using sensors and other technologies are more conducive to protecting the privacy of the elderly. They are less invasive and do not require the elderly to operate independently. However, some participants reported that some devices neglected the details of the daily life of elders when they were designed. This hurts the fragile lives of older people.


*“It is truly annoying, when I get up at night to go to the bathroom, this thing (the action monitor) keeps flashing red, and it is blinding when the lights are not on”. (P15, abandon)*



*“The alarm (voice triggered alarm) goes off randomly at times and the sound is very shrill. In order not to let it scare guests coming to the house. I will turn it off secretly when someone comes”. (P20, abandon)*


##### Selective Order Taking

The promotion of smart technology in home care settings is still in the pilot phase. Many companies do not have the sufficient capacity to handle the service needs sensed by the devices or reported by the elderly on their own. Especially in rural areas where healthcare resources are scarce, even when devices are installed, the services ultimately available to the elderly are still not improved.


*“They prefer to respond to services with lower technical requirements because there are more people who can take such orders. I have figured out their model, so I will only use it for daily care”. (P17, adoption)*


Participants reported that the use of smart technologies such as sensors can reduce the risk of injury to the elderly at home, or they can receive timely help in the face of accidents such as falls. However, in meeting the needs of professional nursing, due to the lack of nursing staff, institutions that sign contracts with smart platforms are more willing to accept low-tech services such as dressing changes.

##### Payment for Services Outside of Government Subsidies

Beyond those, although the government covers the cost of the elders’ initial exposure to smart technology, technology is still only an intermediary for them. The existing care technology cannot yet work completely independently of human services, and the elderly will eventually seek the services of professional caregivers. The additional costs incurred are unbearable for some participants.


*“Although there is no charge for these devices, the service charge (like door-to-door service) is 2–3 times the normal rate. The reason they gave was that it is a special needs service”. (P19, abandon)*


#### 3.2.3. Theme 3: Possible Collateral Damage

Concerns about collateral damage in the adoption of technology by the elderly are a long-standing topic. Collateral damage includes not only physical harm but also unintended harm to the elders’ lifestyles and emotions. In particular, compared with the damage—such as life invasion—that may be caused by using traditional technology, the damage caused by smart technology is more hidden, and the possible negative effects are more powerful. Some of these are within the range of what the elderly can tolerate, and they see them as the price to pay for access to convenient services. However, some participants have given up continuing to use technology after thinking about and adopting some alternative methods.

##### Lack of Empathy

Door-to-door services transferred through smart technology have a specific management process. This not only binds the service provider but also involves elderly individuals. People must follow the rules set by the program to complete each service. Some participants shared with us the specific process of services transferred by smart technology.


*“When the nursing staff comes to my home, they first need to take a photo at my door. After he submits the photo, his device (smartphone or PDA) will automatically match his GPS positioning information to confirm whether he arrives. Although there is no need to take photos or video records during the service, the equipment will automatically calculate their stay time in my home according to the location. After the service is completed, I need to confirm and give a comment on the platform”. (P2, adoption)*


It is different from the traditional way of using the signature of the elderly and telephone call back to reflect the authenticity of the service. Everything under smart technology is full of data calculations. To some extent, this may improve the quality of service; however, some participants perceive a lack of basic empathy for smart technology because, as customers, they have no obligation to sacrifice their plans to cooperate with the management process.


*“The system has a set service time, but many services do not take that long. I had to stay home with him (the nursing staff) after it was over. Otherwise, the system would judge this nursing service failed”. (P12, adoption)*



*“After the service, the caregiver would ask me to give them favorable comments on the platform, saying that it affects their income. I will think that if I do not give him a good comment, will he become very bad when he comes to nursing next time?” (P25, abandon)*


##### Loss of Self-Determination

As an important supplement to traditional care services, using smart technology in the home care environment should continue the traditional concept of being “patient-centered”. Unfortunately, some participants shared with us something quite different. They felt that not only the devices but also the caregivers were only following procedures, guidelines, and orders. This makes them unable to participate in the joint decision-making of care action.


*“We did not communicate much, the nursing staff served as i ordered, and when i tried to discuss with them whether i can change the future nursing plan, they coldly refuse me and tell me to go to the institution to get answers”. (P21, abandon)*


In traditional medical services, healthcare workers often work with the elderly to make treatment plans based on past physical examination reports and real-time inspection data. With the intervention of smart technology, the physical condition of the elderly is constantly recorded. The data used to protect the elderly’s ability to live safely at home turned into cold evidence. The elderly lost their qualification to participate in the nursing plan altogether because their physical data proved what services they would need.

##### Possible Unintended Harm

Devices such as motion sensors are called smart technology (devices). However, it does not predict or understand human self-action. It can only follow the settings and perform its job faithfully. In some specific situations, it may cause unintended harm to elderly individuals.


*“It reminds me to take my pills at 9:00 every day, but sometimes I remember to take it myself and then wonder if I have taken it after hearing the reminder. There were times I thought I had taken pills twice”. (P23, abandon)*


##### Weakens Social Connections

Almost all the participants expressed that they cherish the social relations they have. They worry that they will be cut off from their old social networks because of the long-term use of smart technologies.

In the interview, we found that many elders used to acquire help from fixed doctors when they went to the hospital. With the intervention of smart technology, although the elderly can reduce the frequency of going to the hospital, it also reduces their contact with trusted doctors. They are worried that this will weaken their social connection.


*“If I keep using these technologies and do not go to the hospital, I will lose contact with the doctors and nurses I am familiar with. When I go to the hospital again, I may not be able to get convenient services from them”. (P14, abandon)*


In addition, the information collected and processed by smart technology will also be sent to the families of the elderly through the platform. They can remotely check the physical condition and action track of the elderly. This makes elders worry about whether it will reduce the chances of face-to-face contact with their families after using technology.


*“Honestly, I worry that after using these technologies for a long time, my family members will stop coming over to see me as often as they can because they are at ease enough with me”. (P22, abandon)*


## 4. Discussion

In this study, we discussed the long-term adoption or abandonment of smart technology by the elderly in the home care environment and identified three main aspects that the elderly should consider when considering the future use of technology: direct effectiveness, long-term usability, and the possible collateral damage of smart technology.

### 4.1. Factors Affecting the Experience of the Direct Effectiveness of Smart Technology

Improving the home care environment for our elders differs from the functionalist goal of designers and purchasers who choose smart technology [42]. The elderly, as end users, hold a simpler, more straightforward, pragmatic attitude toward smart technology. They hope that technology can directly meet some unmet needs and has intuitive advantages over traditional methods. The standardization of smart technology in terms of management and processes creates a service atmosphere full of security for elders [19]. It also enables elders to access professional services in the home from traditional healthcare providers that elders trust more [43]. This not only meets their subjective expectations of technology’s advantages but also provides elders with a more convenient and effective experience than traditional services [44]. At the same time, smart technology can help elders manage their health and prevent some risks before problems occur or become more serious [37,45]. This allows elders to better cope with the aging process and have a healthier, more independent living experience [46].

An important reason in the effectiveness category that leads elders to abandon the use of technology is the mismatch between the theory and application of technology. Influenced by the concept of immediate response at the theoretical level of technology, elders subjectively believe that staff and ambulances will appear once they call for service or press a button [47]. However, in reality, this requires secondary matching of staff, and the speed of the response is related to the amount of demand and the number of available caregivers [42]. If they perceive or experience the ineffectiveness of this technology in actual use, they will adopt what they perceive to be a more trustworthy solution. In the future, we need to increase training for the platform staff, optimize caregiver dispatch models, and enhance the responsiveness of the service.

### 4.2. Factors Affecting the Experience of the Long-Term Usability of Smart Technology

In contrast to the previous studies that found that the digital divide and personal intentions influenced the adoption of technology by elders, we reconfirm that some current technologies are already within the range of what elders can use or, at least, are willing to use [43]. This is partly related to the lowering of the barriers to technology use but also to the explosive adoption of information technologies by older people in recent years, such as social media and e-government [48]. On this basis, we should pay more attention to technical standardization and detailed considerations. Current technology devices lack standardization in their operational design, and even devices developed by the same company are completely different, which invariably increases the learning cost for elders. At the same time, researchers often conduct research in manually controlled research environments; they rarely examine specific service access or connectivity issues [37]. Some elders living in rural or medically under-resourced settings have not improved their access to services with the use of technology.

In addition, technology should also try to avoid being invasive in daily life [49]. Sharp alarms and constantly flashing lights not only affect the daily lives of elders but also constantly suggest that they are different from other population groups. These factors reduce the elders’ evaluation of technology usability and hinder their use of technology in the home care environment, and more attention needs to be paid to humanistic care in the design of technology to reduce the intrusiveness into elders’ lives. In addition, although China has made technology accessible to elders through government purchases, there is a lack of uniform standards and diverse means of payment (e.g., health insurance) for the price of follow-up services. In line with previous research findings [6,50], this also hurts the long-term adoption of technology by elders. It is necessary to learn from Japan, Germany, and other countries to incorporate home care into long-term care insurance so as to better improve payment options for the elderly.

### 4.3. Factors Affecting the Experience of the Possible Collateral Damage of Smart Technology

The elders’ perceptions of possible collateral damage from technology adoption continue throughout their technology use. This negative perception (or experience) does not necessarily force elders to abandon the technology outright, but it can affect their positive evaluation of the technology’s effectiveness and usability [37]. Most previous technology adoption studies have been conducted in controlled settings, such as the laboratory, and elders, more often than not, simply report their attitudes toward technology based on the information given [51], ignoring the harm that may occur in specific user settings. For example, in the home care environment, caregivers who come to the home are often not given complete information about the older person, and most caregivers also provide care alone [6]. Under this premise, the lack of individual competence, professional equipment, and team support among different staff members can easily produce uneven service quality, leaving elders concerned about the overall quality of technical services [52].

In addition, the desire of elders to self-determine and participate together in their health management is ignored by professionals and smart devices during the service process under the influence of traditional one-way authoritative approaches [53] and service process constraints [54]. This experience, while not causing actual physical or physiological damage to elders, severely hurts them emotionally. Some studies have shown that verbal and behavioral empathy in health care behaviors is 50% important for service outcomes, while personnel professionalism accounts for only 15–20% [55]. We need to consider the timely introduction of norms for care services based on smart technology platforms and the establishment of quality control systems. In addition, the most specific concern some older people share is about the damage of the long-term use of technology on social networks. Influenced by traditional culture and past life experiences, Chinese seniors believe that care and attention from their children are irreplaceable. They have a strong emotional dependence on and attachment to their children and are concerned that smart technology brings too much imagery of independent living for elders. This concern leads to technology abandonment behavior that requires further consideration of the cultural characteristics of users in technology diffusion. Studies have demonstrated that moderate child involvement facilitates technology adoption behavior and can prevent technology use by elders who do not follow the guidelines [17].

## 5. Conclusions

This study explores the reasons for the long-term adoption and abandonment of smart technology by the elderly in the home care environment. Our research shows that whether the elderly adopt smart technology depends on the user’s experience. Even if the elderly adopt smart technology on the advice of experts or family members, they will give up their previous adoption decision if there is no obvious advantage compared to the traditional way. The main reason why they give up using smart technology is that direct effectiveness is not high, long-term usability is insufficient, and the possibility of collateral damage. In fact, whether smart technology can be applied continuously is not because it is the most advanced technology but because it can meet the needs of the elderly for a long time. Almost all the reasons that lead the elderly to give up using smart technology come from ignoring their living habits and actual needs. In traditional technology, there are many types of individual devices, and their defects not only have little influence on the elderly but are also covered up by the positive results after use. With the addition of smart technology, big data and platforms have integrated these devices and services into a whole package. Small defects originally solved by a single solution will be amplified in the linkage between devices. This requires us to improve the device in real-time through technical means and to change the original function-oriented logic based on the product’s characteristics into holistic thinking based on the needs of the elderly. In addition, for more elders to use smart technology in the home care environment, the government needs to further establish a quality supervision system for care services and improve the means of payment for services; technology developers need to improve technology development norms and reduce the intrusion of technology into daily life; nursing institutions need to strengthen their service’s response speed and promote the rational allocation of regional resources.

There are several limitations to this study. First, the study participants were geographically limited, all from the same region, and may not be as representative in other regions. For example, there are natural differences among the elderly’s technology use abilities and habits in different regions. It will lead to completely different expectations and tolerance for the use of technology. Second, because this study adopts the directed content analysis method, the data collected are relatively limited. This leads to a relative consistency between our findings and the theory suggested by Golant. However, compared with the theoretical framework derived from the literature, the information obtained in this study is closer to the actual reality of smart technology being used by the elderly. In addition, the participants covered only elders in home care environments and did not include other essential stakeholders, such as family members, caregivers, technology developers, and policymakers. There is a need to further explore the reasons for long-term use or abandonment of smart technologies for elders from the perspective of more subjects in future studies.

## Figures and Tables

**Table 1 healthcare-11-02440-t001:** Demographics of participants.

Characteristics	Participant (*n* = 26)
**Gender, *n* (%)**	
Male	15 (58)
Female	11 (42)
**Age (years), mean (SD)**	71.2 (6.7)
**Education status, *n* (%)**	
Primary school and below	5 (19)
Middle school	12 (46)
High school	6 (23)
Higher education and above	3 (12)
**The need for home care services, *n* (%)**	
Daily life care	6 (23)
Professional care	11 (42)
Life or health monitoring	9 (35)
**Group, *n* (%)**	
Long-term adoption group	12 (46)
Abandonment group	14 (54)
**Ways to learn about smart technology, *n* (%)**	
Internet information	3 (12)
Advice from healthcare staff	10 (38)
Publicity in community center	5 (19)
Family members and friends	8 (31)
**Types of smart technologies used, *n***	
Smart eldercare platform	26
Emergency button	20
Wearable or IoT devices	17
Partner or nursing robots	5

## Data Availability

The datasets used and analyzed during this study are available from the corresponding author upon reasonable request.

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
