# Peer review of "Long-Term Adoption or Abandonment of Smart Technology in the Chinese Elderly Home Care Environment: A Qualitative Research Study"

_healthcare, 2023, doi:10.3390/healthcare11172440_

Round 1

Reviewer 1 Report

First of all, thanks for inviting me to review this paper. The paper is interesting to read, and provides extremely valuable insights about elderly’s adoption and abandonment of smart technology in China. One of the concerns I have is that authors seem to have mixed “technology” in general and “smart technology”. Smart technology itself is already a very broad area, covering from smart house, IoT to wearable devices, and the definitions of this term has evolved a lot due to the advancement of technology. **Now focuses more on big data for example. In some texts and references, authors do not differentiate between the two of them. For instance, in lines 60-63, the texts say smart technology but the references, i.e., 13 & 14 are related to health-related Internet use and mobile-based. Not all Internet, or general technology use can be classified as “smart technology”.

In addition, I suggest the paper to be proof-read to improve its flow and readability. Because of these, I think the authors have to do quite a lot of changes to the manuscript and therefore, most of my comments are more general than line-by-line reviewing and commenting.

Introduction. Here, one major improvement that can be done is to provide reliable definitions of smart technologies that the authors refer to in this study. Line 29, I am not too sure if the claim is still valid. but according to https://www.worlddata.info/the-largest-countries.php, the authors might be wrong; suggest to make more moderated claims – like one of the most populous countries. The same goes to line 41. Does it need to be so strong and specific statement “since 2017”? Line 45-47 need reference, as authors claim “a large number of Chinese studies”?

Theory framework. Not sure if this is really required as per journal’s requirement. But I think this part can be actually be placed under research design (Methods).

Methods. First paragraph in 3.3 is not part of data collection. You may move it to research design. Or in a separated sub-section called pre-data collection. In line 153, the authors say they used a directed content analysis method. Is “directed content analysis” a method introduced by “reference 31”? I can’t find this method, but only inductive and deductive content analysis. COREQ in line 169 needs both references and its full name.

Results. The results lack details and contexts for readers to understand. Smart technology is a very broad term and therefore it is difficult for readers to understand what kind of smart technology the participants were referring to. For instance, in 4.2.1.1, what services? What order? In 4.2.1.2, how did P3 use smart technology to choose hospital? In 4.2.1.4, what are the technology intermediaries, and communication and service models here? Is it possible for the authors to describe the devices and/or functionalities in the smart technology they used? I see that they have demonstrated a bit of it via direct quote from P23. However,  that was just one of the some elderly people (which means only a particular device) the authors refer to.

In addition to giving necessary descriptions or contexts, the authors need to make sure that they do not mix their findings with others’ work, or make sure the findings are presented in a clearer way that “this is based on their participants”. For instance, in 4.2.2.1, “In the traditional home care environment, caregivers hired by the elderly and their families usually lack standardization in their workflow”, is this a finding of theirs, or other relevant works? Same for 4.2.2.2. Plus, is this really a selling point? Who claims that? How about being intrusive and invasive? On the other hand, 4.2.3.2 is a good way how authors clearly distinguish between theirs and others’ findings.

In 4.1, is it a coincidence that “males comprised approximately half (n=15) with a mean age of 71 (SD=6.7)”, and the statistics in table 1 (71.2 for age and 6.7 for mean) are the same?) the latter should refer to the entire group of participants, right? In addition, the text in 4.1 with amount of participants do not seem to be coherent with numbers shown in table 1 (the need for home care services, groups for long-term adoption and abandonment group). For instance, in text: “9 participants gave up the use ... ” while in the table is 14? In this sub-section, the authors can report what kind of smart technologies have the participants used. I see this is the second question in their interview outline. By reporting this, we readers can understand better when it comes to the detailed contexts and their usage (my earlier comment).

In 4.2.1.1, what do the authors mean by “mutual concessions”, which have any kind of associations with the use of smart technology? We need more explanations, or contexts here. In 4.2.1.2, is there any special reason why the authors provided both Chinese term and English term of transient ischemic attack? Is the Chinese version slightly different or? In 4.2.2.2, the term “senseless monitoring”. I have checked this term on Google Scholar, and it seems like this is not a common term to be used. Do the authors mean something like “ubiquitous monitoring”? In the quote of P20, precisely what alarm is that?

Theme and subtheme in 4.2.3. I think it is important that the authors draw a clear line on how the potential consequences are related to smart technology, or technology in general. For instance, in 4.2.3.1, those door-to-door services might also be “ordered” via non-smart technologies? In 4.2.3.1, is this a consequence of using smart technology, or just home care being digitalized? In quote from P12, who is “he”, and what is “his service”? In 4.2.3.2, quote from P21, what is the “next step in my care”? (Again, all these are lack of context and necessary explanation). These detailed findings can provide us readers very valuable insights.

The subtheme in 4.2.3.4 does not seem “correct” to me. The social networks are still there when people use technology and are on digital platform. What really was missing is the physical face-to-face human contact. Otherwise, authors need to be more concrete in describing how social contacts are reduced due to the use of smart technologies, i.e., do not need doctor or nurses or any caregivers, or family stop visiting and cut off contact… etc.

Discussion. This section is well-written. However, do double check if relevant works cited here, and their text are “aligned”; refer to my comment about making clear distinctions between “technology” and “smart technology”. The authors have done a great job relating to relevant works but do consider dividing the entire section now into sub-sections with descriptive sub-headings, and probably in separated paragraphs for all the texts in this section now (they are way too long) to improve readability. In line 319, it should be usability not availability. Limitations can be moved to Conclusions as conclusions are pretty short now. Another limitation that I see is that the data collected is more structured but at the same time, more “limited” (see lines 151-152). This has contributed to their findings are not much different from the theory suggested by Golant. Hence, this make the scientific contribution less significant.

Conclusions. The section on conclusions is well supported by the findings although it is a bit short at this moment. I would hence advise the authors to further develop this section by adding more information or details on what implications the findings have for the relevant research fields. Reflect more specifically on “smart technology” and not just technology in general. Some of the components with smart technology is big data, artificial intelligence, machine learning, etc… The authors need to show how all these can be associated to the experiences of elderly (as mentioned earlier, this part with definition and components have to be improved starting from Introduction, and therefore might result in some changes to results and discussions too). This amendment will truly shape a better, clearer implications to policy makers, researchers, and other stakeholders.

Some sentences are too long, too complicated and unclear. Some tenses are wrong too.

Author Response

Dear Reviewer:

We feel great thanks for your professional review work on our manuscript entitled “Long-term adoption or abandonment of smart technology in the Chinese elderly home care environment: a qualitative research”(ID: healthcare-2566839). Those comments are all valuable and very helpful for revising and improving our paper, as well as the important guiding significance to our research. We have studied the comments carefully and have made corrections which we hope meet with approval. Revised portions are marked in red on the resubmitted paper. The main corrections in the paper and the responses to the reviewer’s comments are as follows:

Response to the reviewer#1’s comments:

Comments about the introduction section:

Comment 1:

Here, one major improvement that can be done is to provide reliable definitions of smart technologies that the authors refer to in this study.

Response to comment 1:

We thank the reviewer for the valuable comments. In the newly submitted manuscript, we have summarized the definition of smart technology in the existing literature and given the corresponding explanation.

The corresponding new content is added in lines 52 to 67, page 2.

Comment 2:

Line 29, I am not too sure if the claim is still valid. but according to https://www.worlddata.info/the-largest-countries.php, the authors might be wrong; suggest to make more moderated claims – like one of the most populous countries. The same goes to line 41. Does it need to be so strong and specific statement “since 2017”? Line 45-47 need reference, as authors claim “a large number of Chinese studies”?

Response to comment 2:

We thank the reviewer for the valuable comments. These mistakes you mentioned have been corrected in the new manuscript.

The corresponding new content is added in line 29 & lines 43 to 44, page 1; lines 50, page 2.

Comments about the methods section:

Comment 1:

Theory framework. Not sure if this is really required as per journal’s requirement. But I think this part can be actually be placed under research design (Methods).

Response to comment 1:

We thank the reviewer for the valuable comments. We have placed this part under the research design.

The corresponding new content is added in lines 121 to 133, page 3.

Comment 2:

First paragraph in 3.3 is not part of data collection. You may move it to research design. Or in a separated sub-section called pre-data collection.

Response to comment 2:

We thank the reviewer for the valuable comments. We have moved it in a separated sub-section.

The corresponding new content is added in lines 154 to 160, page 4.

Comment 3:

In line 153, the authors say they used a directed content analysis method. Is “directed content analysis” a method introduced by “reference 31”? I can’t find this method, but only inductive and deductive content analysis.

Response to comment 3:

The directed content analysis method is introduced by ref.31(now is ref.39). In addition to this document, some papers have appeared in recent years to introduce this method. We have attached references that may be useful.

Assarroudi A, Heshmati Nabavi F, Armat MR, Ebadi A, Vaismoradi M. Directed qualitative content analysis: the description and elaboration of its underpinning methods and data analysis process. Journal of Research in Nursing. 2018;23(1):42-55. doi:10.1177/1744987117741667

Comment 4:

COREQ in line 169 needs both references and its full name.

Response to comment 4:

We thank the reviewer for the valuable comments. We have added the full name and reference of COREQ checklist.

The corresponding new content is added in lines 195 to 197, page 4; reference 40, page 14.

Comments about the results section:

Comment 1:

In 4.1, is it a coincidence that “males comprised approximately half (n=15) with a mean age of 71 (SD=6.7)”, and the statistics in table 1 (71.2 for age and 6.7 for mean) are the same?) the latter should refer to the entire group of participants, right? In addition, the text in 4.1 with amount of participants do not seem to be coherent with numbers shown in table 1 (the need for home care services, groups for long-term adoption and abandonment group). For instance, in text: “9 participants gave up the use ... ” while in the table is 14? In this sub-section, the authors can report what kind of smart technologies have the participants used. I see this is the second question in their interview outline. By reporting this, we readers can understand better when it comes to the detailed contexts and their usage (my earlier comment).

Response to comment 1:

We thank the reviewer for this comment and feel sorry for making such a low-level mistake. We have corrected these mistakes in the newly submitted manuscript. In addition, according to your suggestion, we further reported the technology usage of the participants.

The corresponding new content is added in lines 201 to 217, page 5.

Comment 2:

The results lack details and contexts for readers to understand. Smart technology is a very broad term and therefore it is difficult for readers to understand what kind of smart technology the participants were referring to.

Response to comment 2:

We thank the reviewer for the valuable comments. We think this is an excellent suggestion. As you pointed out, we did not give the necessary explanation and context to help readers better understand our findings, which was an oversight on our part. In the newly submitted manuscript, we have added a lot of details about the smart devices used by the participants to help readers better understand our manuscript.

The corresponding new content is added in lines 230 to 238, 242 to 246, 250 to 254, page 6; lines 269 to 279, 297 to 302, 306 to 315, page 7; lines 330 to 334, 347 to 350, 359 to 366, page 8; lines 367 to 373, 384 to 392, 405 to 408, 412 to 415, page 9.

Comment 3:

In addition to giving necessary descriptions or contexts, the authors need to make sure that they do not mix their findings with others’ work, or make sure the findings are presented in a clearer way that “this is based on their participants”. For instance, in 4.2.2.1, “In the traditional home care environment, caregivers hired by the elderly and their families usually lack standardization in their workflow”, is this a finding of theirs, or other relevant works? Same for 4.2.2.2. Plus, is this really a selling point? Who claims that? How about being intrusive and invasive? On the other hand, 4.2.3.2 is a good way how authors clearly distinguish between theirs and others’ findings.

Response to comment 3:

We thank the reviewer for the valuable comments. We have revised the contents of the manuscript that may cause readers to misunderstand.

The corresponding new content is added in lines 230 to 238,page 6; lines 306 to 315, page 7.

Comments about the discussion section:

Comment 1:

The authors have done a great job relating to relevant works but do consider dividing the entire section now into sub-sections with descriptive sub-headings, and probably in separated paragraphs for all the texts in this section now (they are way too long) to improve readability.

Response to comment 1:

We thank the reviewer for the valuable comments. We have divided this section into sub-sections with descriptive sub-headings. At the same time, we separate paragraphs for all the text to improve readability.

Comment 2:

Limitations can be moved to Conclusions as conclusions are pretty short now.Another limitation that I see is that the data collected is more structured but at the same time, more “limited” (see lines 151-152). This has contributed to their findings are not much different from the theory suggested by Golant. Hence, this make the scientific contribution less significant.

Response to comment 2:

We thank the reviewer for the valuable comments. We have moved this part to conclusions and add the limitation you mentioned.

The corresponding new content is added in lines 534 to 538,page 12.

Comments about the conclusion section:

Comment 1:

I would hence advise the authors to further develop this section by adding more information or details on what implications the findings have for the relevant research fields.

Response to comment 1:

We thank the reviewer for the valuable comments. In the newly submitted manuscript, we further enriched our conclusions.

The corresponding new content is added in lines 507 to 523,pages 11 to 12.

We tried our best to improve the manuscript and made some changes in the manuscript. And here we did not list the changes but marked them in red in the revised paper. We appreciate for reviewer’s warm work earnestly and hope that the correction will meet with approval. If there are any other modifications we could make, we would like very much to modify them and appreciate your help.

Once again, thank you very much for your comments and suggestions.

With best regards,

Authors

Reviewer 2 Report

This is qualitative research paper that has aimed to study the adoption and abandonment of at home smart technology for the elderly in Nanjing, Eastern China. The study was done by 3 researchers, each covering a different part of the methodology, while evaluating the final results together. An interview with two sets of questions took place, one for participants who had adopted smart technology in a home care environment, and the other who had not. Moreover, there were set requirements for the participant cohort, they were all made aware of the conditions of the study, and all signed a consent form for their results to be used for the purpose of the study. There was a total of 26 participants. The outcome of the research showed that the usage of smart technology in a home care environment was dependent on the user experience, effective usage of the devices as well as the long-term impact of technology on their personal and social lives. 

Strengths 

1.     Presentation of the research paper was clear and concise with the use of relevant titles and appropriate subtitles to make it easy for the reader to locate specific points. 

2.     Theory Framework: Well-explained and related to the research in discussion in lines 97-99. 

3.     Referencing: A good range of references, including very recent 2022, but also dating back to 1997. 

4.     Methods: The research paper is repeatable as all the conditions of the participants for the study were stated. The interview questions were also given with the manuscript. Research design, sampling and the data collection process/analysis was provided in detail for ease of repeatability. 

5.     A signed consent was taken from all participants so there are no ethical issues.

Corrections and Comments

1.     Between line 37-39, the author writes about the traditions of Chinese families, however there is no reference for this. 

2.     Between line 76-80, this sentence is too long and needs be shortened or turned into two sentences. For example, on line 78 after “adoption of technology by the elderly” there should be a full stop and then start a new sentence.  

3.     Between line 80-82, the sentence is missing come key grammatical filler words. For example, “However, the essence of intelligent technology in a home care environment is to put the elderly at the centre of the care network”. 

4.     Between line 85-86, sentence needs to be worded better as the point is not clear. 

5.     Method: The author writes that all participants were made aware of the purpose, method and usage of the research. The author may want to comment in the discussion whether this could have had an effect on the results obtained and if not, mention the reason why it did not. 

6.     This study was done in Nanjing, Eastern China. This area may use less technology in general compared to other areas in China, which may also affect the results obtained. For example, the elderly in central China might use more smart technology due to the general population in central China using more technology. This limitation is touched upon in line 399 however it can be discussed in more detail in the discussion.  

7.     On line 142, the author writes about the researcher assessing the participants body language and facial expressions during the interviews. However, this was not portrayed in the results in any way. For example, there could have been a graph of a table showing positive and negative physical expressions of the participants and correspond the results to whether they had abandoned or adopted long term smart technology. 

8.     Results: The raw results (answers to the interview questions) were not fully given. Specific sections of the raw results were elaborated on in the results but a table with all the results may have been good and useful for the reader. 

9.     The sentence between line 351-353 does not have a reference. 

10.  The sentence between line 364-366 lacks punctuation and may be shorted. 

English is fine, however some sentences need to be reworded and/or shortened for the reader's ease. This is explained in the review. 

Author Response

Dear Reviewer:

We feel great thanks for your professional review work on our manuscript entitled “Long-term adoption or abandonment of smart technology in the Chinese elderly home care environment: a qualitative research”(ID: healthcare-2566839). Those comments are all valuable and very helpful for revising and improving our paper, as well as the important guiding significance to our research. We have studied the comments carefully and have made corrections which we hope meet with approval. Revised portions are marked in red on the resubmitted paper. The main corrections in the paper and the responses to the reviewer’s comments are as follows:

Response to the reviewer#2’s comments:

Comment 1: 

Between line 37-39, the author writes about the traditions of Chinese families, however there is no reference for this. 

Response to comment 1: 

We thank the reviewer for the valuable comment. We have added this part to the manuscript.

The corresponding new content is added in lines 37 to 40, page 1.

Comment 2:

Between line 76-80, this sentence is too long and needs be shortened or turned into two sentences. For example, on line 78 after “adoption of technology by the elderly” there should be a full stop and then start a new sentence.

Response to comment 2:

We thank the reviewer for pointing this out. We have revised this sentence in the manuscript.

The corresponding new content is added in lines 95 to 99, page 2 to 3.

Comment 3: 

Between line 80-82, the sentence is missing come key grammatical filler words. For example, “However, the essence of intelligent technology in a home care environment is to put the elderly at the centre of the care network”. 

Response to comment 3:

We thank the reviewer for pointing this out. We have revised this sentence in the manuscript.

The corresponding new content is added in lines 100 to 101, page 3.

Comment 4: 

Between line 85-86, sentence needs to be worded better as the point is not clear. 

Response to comment 4:

We thank the reviewer for pointing this out. We have revised this sentence in the manuscript.

The corresponding new content is added in lines 104 to 106, page 3.

Comment 5: 

Method: The author writes that all participants were made aware of the purpose, method and usage of the research. The author may want to comment in the discussion whether this could have had an effect on the results obtained and if not, mention the reason why it did not. 

Response to comment 5:

We thank the reviewer for pointing this out. The reason why we write this sentence is to show that we fully respect and guarantee the right to know of all participants. We have rewritten this sentence in the manuscript.

The corresponding new content is added in lines 115 to 117, page 3.

Comment 6: 

This study was done in Nanjing, Eastern China. This area may use less technology in general compared to other areas in China, which may also affect the results obtained. For example, the elderly in central China might use more smart technology due to the general population in central China using more technology. This limitation is touched upon in line 399 however it can be discussed in more detail in the discussion.  

Response to comment 6:

We thank the reviewer for pointing this out. We have discussed this issue further in the limitation section.

The corresponding new content is added in lines 532 to 534, page 12.

Comment 7: 

On line 142, the author writes about the researcher assessing the participants body language and facial expressions during the interviews. However, this was not portrayed in the results in any way. For example, there could have been a graph of a table showing positive and negative physical expressions of the participants and correspond the results to whether they had abandoned or adopted long term smart technology. 

Response to comment 7:

We thank the reviewer for pointing this out. We tried to add the participants' body language and facial expressions to the study when writing the first draft of the manuscript. However, we found that participants' body language and facial expressions were not well related to their final behavior. The smart technology is not perfect at present, which leads each participant to have a good and bad experience of it. When recalling the advantages of the technology they used, they showed an expression of excitement and joy. When they think of the troubles this technology has brought them, they obviously show a depressed attitude. At the same time, each participant has different tolerance for technology. Even though they show frustration and dissatisfaction, they believe that these technologies will improve. They are willing to continue to use it. Therefore, in order not to confuse readers with our manuscript, we deleted this sentence in the method part.

Comment 8: 

The raw results (answers to the interview questions) were not fully given. Specific sections of the raw results were elaborated on in the results but a table with all the results may have been good and useful for the reader. 

Response to comment 8:

We thank the reviewer for pointing this out. In the newly submitted manuscript, we added some raw results to interview questions. For example, the way to learn about smart technology and the type of smart technology they used. Since other interview topics are more about participants' reports on the experience of using smart technology, the raw results with commonality have been explained in the results. Affected by personal subjective factors, the rest of the raw results are relatively messy. Many answers only appear once and cannot be well classified into a certain category. Therefore, it is difficult for us to show them all in the form of tables.

The corresponding new content is added in lines 209 to 217 and table 1, page 5.

Comment 9: 

The sentence between line 351-353 does not have a reference.

Response to comment 9:

We thank the reviewer for pointing this out. We have added references to this section.

Comment 10: 

The sentence between line 364-366 lacks punctuation and may be shorted.

Response to comment 10:

We thank the reviewer for pointing this out. We have revised this sentence in the manuscript.

The corresponding new content is added in lines 471 to 473, page 11.

We tried our best to improve the manuscript and made some changes in the manuscript. And here we did not list the changes but marked them in red in the revised paper. We appreciate for reviewer’s warm work earnestly and hope that the correction will meet with approval. If there are any other modifications we could make, we would like very much to modify them and appreciate your help.

Once again, thank you very much for your comments and suggestions.

With best regards,

Authors
